# A *cis*-acting bidirectional transcription switch controls sexual dimorphism in the liverwort

Tetsuya Hisanaga[1],[†] , Keitaro Okahashi[2],[†] , Shohei Yamaoka[2] , Tomoaki Kajiwara[2] ,
Ryuichi Nishihama[2] , Masaki Shimamura[3] , Katsuyuki T Yamato[4] , John L Bowman[5] ,
Takayuki Kohchi[2],* & Keiji Nakajima[1],**

## Abstract

Plant life cycles alternate between haploid gametophytes and diploid sporophytes. While regulatory factors determining male and female sexual morphologies have been identified for sporophytic reproductive organs, such as stamens and pistils of angiosperms, those regulating sex-specific traits in the haploid gametophytes that produce male and female gametes and hence are central to plant sexual reproduction are poorly understood. Here, we identified a MYB-type transcription factor, MpFGMYB, as a key regulator of female sexual differentiation in the haploid-dominant dioicous liverwort, *Marchantia polymorpha*. MpFGMYB is specifically expressed in females and its loss resulted in female-to-male sex conversion. Strikingly, Mp*FGMYB* expression is suppressed in males by a *cis*-acting antisense gene *SUF* at the same locus, and loss-of-function *suf* mutations resulted in male-to-female sex conversion. Thus, the bidirectional transcription module at the Mp*FGMYB/SUF* locus acts as a toggle between female and male sexual differentiation in *M. polymorpha* gametophytes. *Arabidopsis thaliana* Mp*FGMYB* orthologs are known to be expressed in embryo sacs and promote their development. Thus, phylogenetically related MYB transcription factors regulate female gametophyte development across land plants.

**Keywords** antisense transcription; lncRNA; *Marchantia polymorpha*; R2R3 MYB-type transcription factor; sexual differentiation

**Subject Categories** Plant Biology; RNA Biology; Transcription

The EMBO Journal (2019) 38: e100240

See also: **F Berger** (March 2019)

## Introduction

Life cycles in land plants alternate between a haploid gametophyte, in which male and female gametes are produced, and a diploid sporophyte, which produces haploid spores via meiosis. Sexual differentiation in land plants is best characterized in flowering plants, where male- and female-specific organs of the sporophyte produce male and female gametophytes, respectively. Thus, in flowering plants, sexual development occurs sequentially in sporophyte and gametophyte generations. In contrast, in earlier diverging lineages of land plants, i.e., non-seed plants, little or no sexual differentiation is evident in the sporophytic generation, with sexual differentiation occurring essentially exclusively in the gametophyte generation.

In the context of land plants, sexual differentiation has been investigated in flowering plants wherein the specification of male and female sporophytic floral organs, i.e., stamens and pistils, by the ABC genes has been elucidated (Schwarz-Sommer *et al*, 1990; Bowman *et al*, 1991). One plausible hypothesis is that the sexual differentiation of the sporophyte is imposed upon the retained gametophytes developing within the sporophytic tissues. Thus, during land plant evolution, sexual differentiation has shifted from a purely gametophytic program to a situation where the sporophyte controls sexual differentiation of gametophytes. However, it is an open question whether there exist regulators of gametophytic sexual differentiation that are conserved across land plants.

To decipher the mechanisms by which gametophytic sexual differentiation is established in land plants, it is essential to study sexual differentiation in basal lineages. The liverwort *Marchantia polymorpha*, a recently revived model bryophyte, provides a unique opportunity to study sexual differentiation in gametophytes (Bowman, 2016; Shimamura, 2016; Bowman *et al*, 2017). *Marchantia polymorpha* has several attributes facilitating investigation of the genetic regulation of sexual reproduction such as clear sexual

1 Graduate School of Science and Technology, Nara Institute of Science and Technology, Ikoma, Nara, Japan
2 Graduate School of Biostudies, Kyoto University, Kyoto, Japan
3 Graduate School of Science, Hiroshima University, Kagamiyama, Higashi-Hiroshima, Japan
4 Faculty of Biology-Oriented Science and Technology, Kindai University, Kinokawa, Wakayama, Japan
5 School of Biological Sciences, Monash University, Melbourne, Vic., Australia
 *Corresponding author. Tel: +81-75-753-6389; Fax: +81-75-753-6127; E-mail: tkohchi@lif.kyoto-u.ac.jp
 **Corresponding author. Tel: +81-743-72-5560; Fax: +81-743-72-5569; E-mail: k-nakaji@bs.naist.jp
 †These authors contributed equally to this work

dimorphism in the dominant gametophyte phase and asexual propagation through gemmae formation that allows maintenance of gamete-lethal mutants, in addition to general advantages as a model plant species such as available genome sequence and efficient genetic manipulation techniques (Ishizaki *et al*, 2016; Bowman *et al*, 2017; Sugano *et al*, 2018). Taking advantage of these attributes, recent studies utilizing *M. polymorpha* have revealed several key factors controlling critical steps of sexual plant reproduction (Koi *et al*, 2016; Rövekamp *et al*, 2016; Nakajima, 2017; Yamaoka *et al*, 2018).

In the reproductive phase of their haploid-dominant life cycle, *M. polymorpha* plants exhibit sexual dimorphism depending on the presence of either female (X) or male (Y) sex chromosomes (Fig 1; Bowman *et al*, 2017; Shimamura, 2016; Yamato *et al*, 2007; here, we use X and Y, not U and V, according to the convention of liverwort researchers). Female gametophytes form sexual branches with finger-like rays (archegoniophores) at the apical notch region (meristem) of a vegetative structure called the thallus. Female sexual organs (archegonia) develop at the base of each ray, and a single egg cell differentiates in each archegonium (Fig 1A). In a similar manner, male gametophytes form sexual branches with a disk-shaped morphology (antheridiophores), in which male sexual organs (antheridia) develop and eventually produce motile sperm (Fig 1B). While classical genetic studies predict the existence of a dominant "*Feminizer*" on the X chromosome (Haupt, 1932), mechanisms controlling sexual differentiation of *M. polymorpha* are largely unknown. In contrast, in angiosperms with a diploid-dominant life cycle, female and male gametophytes are highly reduced to seven-celled embryo sacs and three-celled pollen grains, respectively, and their sex-specific differentiation is dependent upon the sporophytic generation (Fig 1C).

In this study, we identified a *cis*-acting bidirectional transcription module as a toggle switch between female and male differentiation in *M. polymorpha*. This module consists of Mp*FGMYB*, encoding an ortholog of previously identified regulators of female gametophyte development in *Arabidopsis thaliana*, and its antisense gene *SUF* producing a long non-coding RNA (lncRNA). Our study suggests that members of this MYB subfamily regulate female sexual differentiation in the haploid growth phase of land plants, while their sex-specific expression is likely regulated by divergent inputs.

## Results

### Conserved MYB transcription factors are specifically expressed in the female gametophytes of *Marchantia polymorpha* and *Arabidopsis thaliana*

To identify evolutionarily conserved regulators of female sexual differentiation in land plants, we compared transcriptome datasets of archegonia and thalli of *M. polymorpha* (see Data availability). Genes preferentially expressed in archegonia were screened and further selected for enrichment of related genes in the published transcriptome datasets of *A. thaliana* female gametophytes (embryo sacs; Yu *et al*, 2005; Steffen *et al*, 2007; Wuest *et al*, 2010). Among the 23 genes thus identified (Table EV1), we focused on *Mapoly0001s0061* encoding a MYB-type transcription factor. In a phylogenetic tree constructed from the amino acid sequences of MYB domains, *Mapoly0001s0061* was found to be closely related to three *Arabidopsis* genes, At*MYB64*, At*MYB119*, and At*MYB98*, as well as two homologs in the moss *Physcomitrella patens* (Fig 2A and B). Because our genetic analyses indicated a role of *Mapoly0001s0061* in female gametophyte development in *M. polymorpha* (see below) as do the three *Arabidopsis* homologs in the embryo sac, the female gametophytes of flowering plants (Kasahara *et al*, 2005; Punwani *et al*, 2007; Rabiger & Drews, 2013), we named *Mapoly0001s0061 FEMALE GAMETOPHYTE MYB* (Mp*FGMYB*), following the *Marchantia* nomenclatural guidelines (Bowman *et al*, 2016), and hereafter refer to the clade including these genes as the FGMYB subfamily (Fig 2A).

PCR analyses detected Mp*FGMYB* in both male and female genomic DNAs, indicating the autosomal localization of Mp*FGMYB* (Fig 2C). As expected from the transcriptome data (Bowman *et al*, 2017), Mp*FGMYB* was predominantly expressed in the archegoniophores of female plants and the sporophytes (Fig 2D). Plants harboring a transcriptional Mp*FGMYB* reporter construct (Mp*FGMYBpro: Citrine-NLS:*Mp*FGMYB3′*) exhibited reporter fluorescence in the archegonia (Fig 2E). In a functionally complemented translational reporter line (gMp*FGMYBresist-Citrine*, see below), Citrine fluorescence was localized in the nuclei of rescued archegonia (Fig 2F). As reported previously (Rabiger & Drews, 2013; Waki *et al*, 2013), our transcriptional and translational reporters for At*MYB64* were specifically expressed in embryo sacs of *A. thaliana* (Fig 2G and H; Rabiger & Drews, 2013; Waki *et al*, 2013).

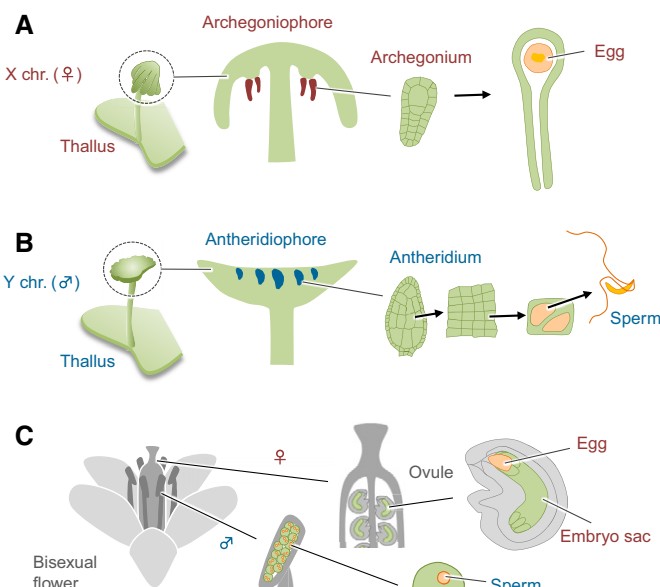

**Figure 1.  Schematic representations of reproductive development in *Marchantia polymorpha* and *Arabidopsis thaliana*.**

A, B   Development of X chromosome-containing female (A) and Y chromosome-containing male (B) *M. polymorpha* plants.

C   Development of embryo sac and pollen, in bisexual flowers of *A. thaliana*.

Data information: In all schemes, gametophytes (haploid) are shown in green, and germline cells in orange. Sporophytic organs (diploid) are shown in gray.

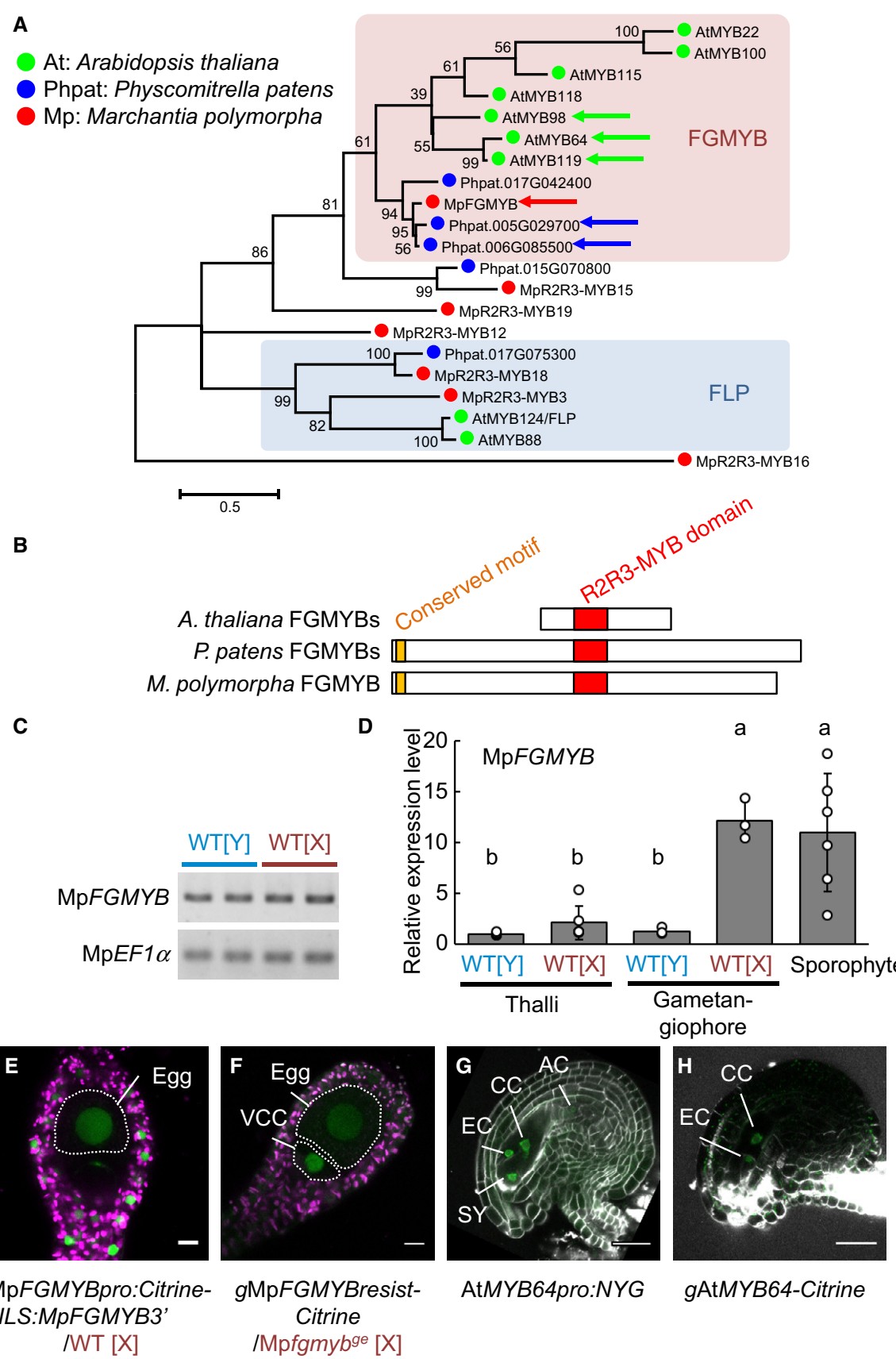

**Figure 2.**

◀

**Figure 2.  FGMYB genes are phylogenetically closely related to each other and expressed in female gametophytes.**

A    Phylogenetic tree of R2R3-MYB proteins of clades 11, 12, and 14–16, as described by Bowman *et al* (2017), from representative land plant species constructed using the maximum-likelihood method based on conserved MYB domain sequences. See Source Data for the sequences used and accession numbers. Numbers at nodes indicate bootstrap values calculated from 1,000 replicates. The tree is drawn to scale, with branch lengths reflecting the number of substitutions per site. Scale bar, 0.5 substitutions per site. Arrows indicate FGMYB orthologs involved in embryo sac development in *Arabidopsis thaliana*, MpFGMYB of *Marchantia polymorpha* (this study), and the most similar *Physcomitrella patens* genes shown in (B). The FGMYB clade is shaded in pink, and a distantly related FOUR LIPS (FLP) clade is in blue.

B    Schematic representations of the FGMYB polypeptide structures. R2R3 MYB domains are shown in red and a conserved amino-terminal motif of PpFGMYBs and MpFGMYB in orange.

C    Genomic PCR analysis indicating the existence of MpFGMYB in both male [Y] and female [X] genomes of *M. polymorpha*. Two biological replicates were analyzed. Autosomal MpEF1α was used as a control.

D    Real-time RT–PCR analyses indicating preferential accumulation of MpFGMYB transcripts in female sexual organs and the sporophytes. MpEF1α was used for normalization. Measurements of six biological replicates for thalli and sporophyte, and three biological replicates for gametangiophore are plotted. Bars represent mean ± SD. Symbols above the bars indicate grouping by $P < 0.05$ in a Tukey–Kramer test.

E    A transcriptional reporter with 5′- and 3′-flanking sequences revealed transcription of MpFGMYB throughout mature archegonia. Scale bar, 10 μm. Magenta, chlorophyll autofluorescence; green, Citrine fluorescence.

F    MpFGMYB-Citrine fusion proteins expressed using the 5′- and 3′-flanking sequences rescued the Mp*fgmyb^ge-1* mutant and accumulated in the nuclei of the egg and the ventral canal cell (VCC; Shimamura, 2016). Scale bar, 10 μm. Magenta, chlorophyll autofluorescence; green, Citrine fluorescence.

G    A transcriptional At*MYB64* reporter (At*MYB64-NLS-YFP-GUS (NYG)*) is specifically expressed in all four cell types of the *A. thaliana* embryo sac (Rabiger & Drews, 2013; Waki *et al*, 2013). Scale bar, 25 μm. Green, YFP fluorescence; white, cell walls.

H    Expression of AtMYB64-Citrine fusion proteins under the At*MYB64* promoter was detected in the central cells (CC) and egg cells (EC) of the *A. thaliana* embryo sac (Rabiger & Drews, 2013). Scale bar, 25 μm. Green, Citrine fluorescence; white, cell walls.

Source data are available online for this figure.

## Loss-of-function Mp*FGMYB* alleles confer a male morphology to female liverworts

Previous studies demonstrated that At*MYB64* and At*MYB119* have critical roles in the development of the embryo sac (Rabiger & Drews, 2013). Similarly, another gene, At*MYB98*, is known to be required for the differentiation and function of the synergids, two of the seven cells constituting the embryo sac (Kasahara *et al*, 2005; Punwani *et al*, 2007). Thus, the preferential expression of Mp*FGMYB* in the liverwort archegonia suggests that the *FGMYB* genes have evolutionarily conserved roles in the development of female gameto-phytes in land plants. To explore this possibility, we generated knockout mutants of Mp*FGMYB* using clustered regularly inter-spaced short palindromic repeats/CRISPR-associated endonuclease 9 (CRISPR/Cas9) technology (Sugano *et al*, 2018). We obtained four independent loss-of-function Mp*fgmyb* lines with insertions or dele-tions that created premature stop codons in the MYB domain-coding region (Fig 3A and Appendix Fig S1A), of which three (Mp*fgmyb-1^ge*, Mp*fgmyb-2^ge*, Mp*fgmyb-6^ge*) were genetically female, while the other (Mp*fgmyb-4^ge*) was genetically male (diagnosed by sex chromosome-linked markers; Fig 3B and Appendix Fig S1C).

The Mp*fgmyb* mutants were morphologically indistinguishable from the wild-type plants during the vegetative growth period (Appendix Fig S2). Genetically male Mp*fgmyb* mutants (hereafter designated Mp*fgmyb* [Y]) were also indistinguishable from wild-type males during reproductive growth (Fig 3F, G, K and L). By contrast, genetically female Mp*fgmyb* mutants (hereafter designated Mp*fgmyb* [X]) exhibited a striking sex conversion phenotype; antheridiophores developed in place of archegoniophores (Figs 3C and D, and EV2A). Furthermore, the antheridiophores of Mp*fgmyb* [X] contained antheridia (Fig 3I, compare with 3H, and Fig EV2B). These phenotypes were rescued by introducing a Mp*FGMYB* genomic fragment containing synonymous mutations to resist the remaining CRISPR/Cas9 activity (gMp*FGMYBresist*) in the mutants (Fig 3E and J, and Appendix Fig S1B), confirming a causal relation-ship between the female-to-male sex conversion phenotype of Mp*fgmyb* [X] and the loss of Mp*FGMYB* function.

The sex conversion phenotype of Mp*fgmyb* [X] could also be rescued by expressing MpFGMYB-Citrine fusion proteins under the same regulatory sequences as those used in the transcriptional reporter lines (gMp*FGMYBresist-Citrine*; Fig 2F). This line exhibited no Citrine florescence in the apical notch region of vegetative thalli (Fig EV1A and B). After induction of reproductive growth by far-red (FR) light treatment (Ishizaki *et al*, 2016), MpFGMYB-Citrine proteins accumulate in ventral apical notch regions where archego-niophores will develop (Fig EV1A and C), and later in developing archegoniophores (Fig EV1A and D), consistent with the requirement of Mp*FGMYB* in female sexual differentiation of *M. polymorpha*.

## Mp*fgmyb* mutant females produce sperm with nearly normal morphology but lacking motility

The results presented so far indicate a key role for Mp*FGMYB* in the female sexual differentiation of *M. polymorpha*. In its absence, male sexual differentiation proceeds as the default program. This female-dominant mode of sex differentiation is consistent with the classical observation that rare diploid gametophytes of *M. polymorpha* carry-ing both X and Y chromosomes exhibit a female morphology (Haupt, 1932); however, it was still unclear whether the loss of Mp*FGMYB* function alone was sufficient to generate functional sperm in the absence of the Y chromosome.

To address this question, we performed a histological analysis and found that spermiogenesis proceeds in Mp*fgmyb* [X] essentially as it does in the wild-type and Mp*fgmyb* [Y] antheridia (Fig EV2C–J). Moreover, sperm collected from Mp*fgmyb* [X] antheridia exhib-ited nuclear condensation and flagella formation (Fig 4A–C). Consistently, two autosomal genes implicated in sperm morphogen-esis and known to be specifically expressed in the antheridiophores, *PROTAMINE-LIKE* (Mp*RPM*) and *DYNINE LIGHT CHAIN7* (Mp*LC7*; Higo *et al*, 2016), were expressed in the antheridiophores of Mp*fgmyb* [X] (Fig 4D). Furthermore, while the expression of the archegoniophore-specific autosomal genes was suppressed in Mp*fgmyb* [X], X chromosome-linked genes expressed in vegetative thalli (Bowman *et al*, 2017) were still expressed in Mp*fgmyb* [X]

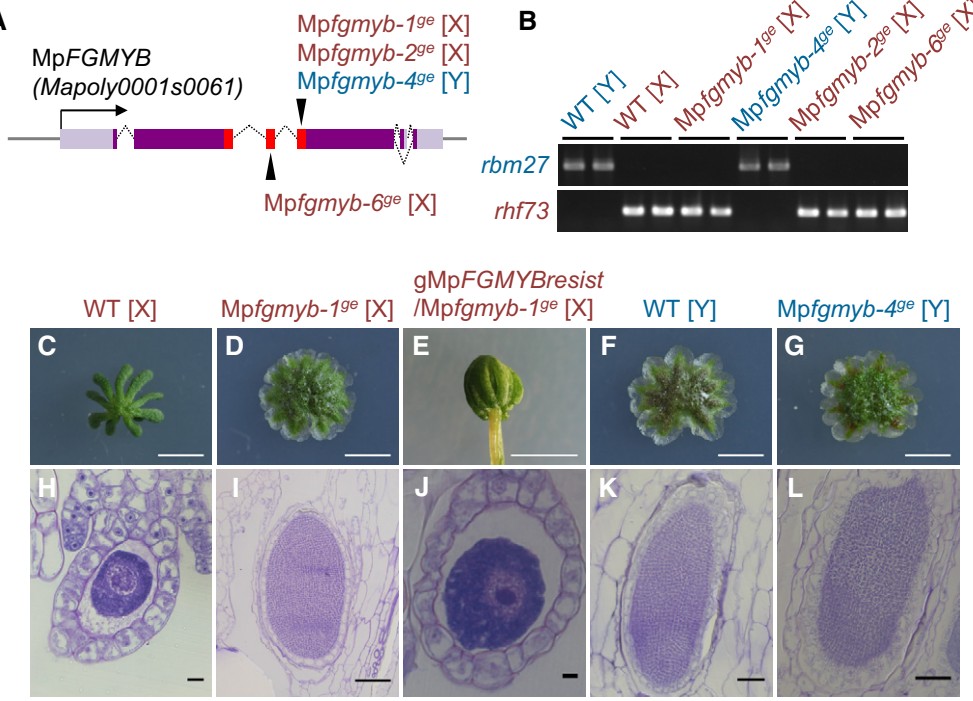

**Figure 3. Loss of MpFGMYB function results in female-to-male conversion.**

A    Mp*FGMYB* gene structure and locations of Mp*fgmyb* mutations. Gray line, 5′- and 3′-flanking sequences; light purple box, UTR; dark purple box, coding region; red box, MYB domain-coding region; arrowheads, mutation positions; black arrow, transcriptional direction; dotted line, splice patterns.

B    Diagnosis of genetic sex using Y chromosome-linked and X chromosome-linked *rbm27* and *rhf73* markers, respectively. Two biological replicates were analyzed for each genotype.

C–L    Gametangiophore morphology (C–G) and gamete development (H–L) of wild-type and mutant plants. Scale bars, 5 mm (C–G), 10 µm (H and J), 100 µm (I, K, and L).

Source data are available online for this figure.

antheridia (Fig EV3A). These data indicate that the feminization capacity of Mp*FGMYB* is primarily associated with the sex-specific expression of autosomal genes involved in sexual differentiation, and not with the expression of sex chromosome-linked genes.

The sperm of Mp*fgmyb* [X] plants exhibited abnormal morphologies, such as incompletely condensed nuclei and short flagella (Fig 4B, compare with 4A and C). Transmission electron microscopy revealed that most Mp*fgmyb* [X] flagella had irregular axonemes, lacking the "9 + 2" arrangement of microtubules (Carothers & Kreitner, 1968) typically seen in wild-type sperm (Fig 4E and F). Consistently, sperm produced in Mp*fgmyb* [X] antheridia were immotile and did not enter wild-type archegonia (Fig EV3B–D). Thus, while the loss of Mp*FGMYB* function resulted in an almost complete female-to-male sex conversion, the formation of functional sperm requires additional factors that are likely encoded by the Y chromosome (Bowman *et al*, 2017).

**Expression of MpFGMYB is suppressed by its *cis*-acting antisense gene SUF in males**

The striking sex conversion phenotype caused by the autosomal Mp*fgmyb* mutations raised the question as to how Mp*FGMYB* expression is tightly suppressed in males. A close inspection of our RNA sequencing data revealed the male-specific accumulation of antisense lncRNAs derived from the Mp*FGMYB* locus (Fig 5A and

B), which we named *SUPPRESSOR OF FEMINIZATION* (*SUF*). Real-time RT–PCR analyses revealed that *SUF* transcripts accumulated in male gametophytes and in sporophytes, while negligible accumulation was detected in female gametophytes. In male gametophytes, antheridiophores accumulated a significantly higher amount of *SUF* transcripts than thalli (Fig EV4A). Importantly, 5′ and 3′ RACE PCR revealed an invariable 5′ end and a polyadenylation site of *SUF* transcripts, indicating that *SUF* constitutes a strictly defined transcription unit of RNA polymerase II (Appendix Fig S3). Strand-specific RT–PCR analyses confirmed *SUF* transcript accumulation in wild-type males both before and after the induction of sexual reproductive growth by far-red irradiation (Fig EV4B).

The mutually exclusive expression patterns of Mp*FGMYB* and *SUF* in gametophytes suggested a possible regulatory mechanism in which *SUF* suppresses Mp*FGMYB* expression in the males, as has been reported for other antisense lncRNAs (Xue *et al*, 2014; Kopp & Mendell, 2018). To test this possibility, we deleted a 1-kb region at the predicted transcription start site (TSS) of *SUF* in the wild-type males using genome editing, without affecting the Mp*FGMYB*-coding sequence (Fig 5A and Appendix Fig S4). As expected, the resulting male mutants (hereafter designated *suf^{ge}* [Y]) lost *SUF* expression and instead gained Mp*FGMYB* expression after the induction of reproductive growth by far-red irradiation (Fig 5B). Furthermore, *suf^{ge}* [Y] plants formed archegoniophores and archegonia that expressed autosomal genes whose expression is female-specific in

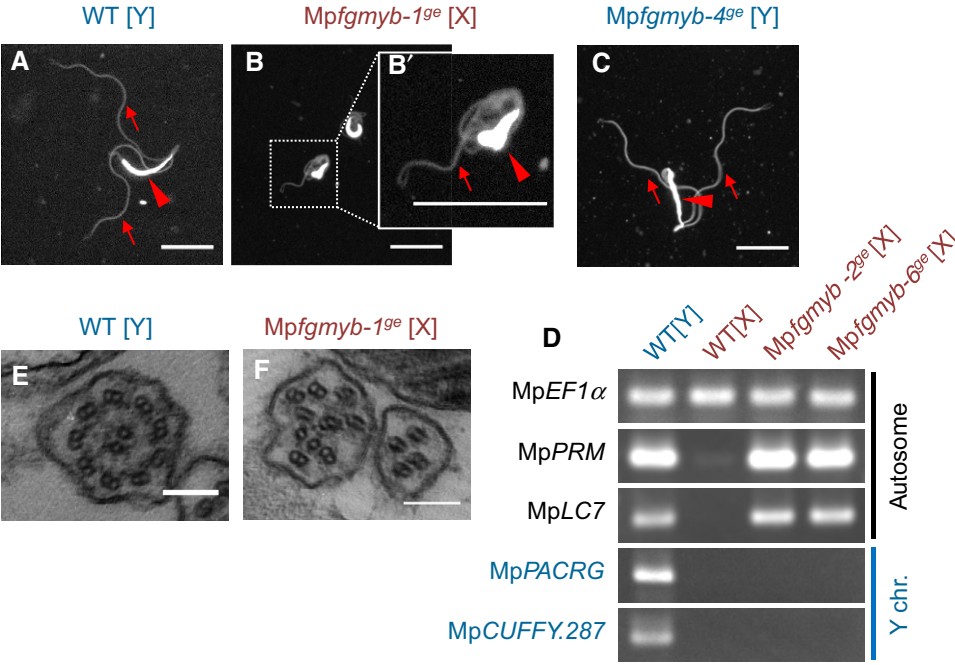

**Figure 4.  Loss of Mp*FGMYB* function results in sperm formation in genetically female plants.**

A–C  DAPI-staining visualization of sperm formation in wild-type male (A), Mp*fgmyb* [X] (B), and Mp*fgmyb* [Y] (C) plants. Note that background DAPI staining visualizes flagella (arrows) in addition to nuclei (arrowheads). (B′) is an enlarged image of the boxed region in (B), visualizing an incompletely condensed nucleus. Scale bar, 5 μm.

D  RT–PCR analysis indicating acquisition of male-like autosomal gene expression patterns in Mp*fgmyb* [X] antheridiophores. Two independent Mp*fgmyb* [X] mutant alleles were analyzed.

E, F  TEM analyses visualizing the abnormal arrangement of axonemal microtubules in Mp*fgmyb* [X] sperm (F), as compared with those of wild-type males (E). Scale bar, 100 nm.

Source data are available online for this figure.

the wild type (Bowman *et al*, 2017), the opposite phenotype to that of Mp*fgmyb* [X] (Figs 5C and D, and EV5). However, archegonia produced in *suf*$^{ge}$ [Y] did not produce egg cells, suggesting that this process may require the function of genes present on the X chromosome (Fig 5D). This male-to-female sex conversion phenotype was suppressed by additional loss-of-function mutations in the MpFGMYB-coding region (*suf*$^{ge}$ Mp*fgmyb*$^{ge}$ [Y]; Figs 5A, E, F, and EV5, and Appendix Fig S4). Furthermore, male plants lacking the entire Mp*FGMYB/SUF* locus (designated *complete deletion* [Y]) developed normal antheridiophores and antheridia expressing the male-specific genes (Figs 5A, G, H, and EV5, and Appendix Fig S4). These data indicate that the male-to-female sex conversion phenotype of *suf* [Y] is caused by the misexpression of Mp*FGMYB*. Taken together, our data support the notion that *SUF* suppresses Mp*FGMYB* expression in males and thereby allows a default male differentiation program to proceed.

To determine whether *SUF* suppresses Mp*FGMYB* expression *in cis* or *in trans*, we made a *SUF* overexpression construct using the strong constitutive Mp*EF1α* promoter (Althoff *et al*, 2014) and introduced it to *suf*$^{ge}$ [Y] in a non-targeted manner. Interestingly, *SUF* overexpression did not rescue the sex conversion phenotype of *suf*$^{ge}$ [Y] (Fig 6A), despite strong accumulation of *SUF* transcripts (Fig 6B). Furthermore, genetically male plants harboring an Mp*FGMYB* transgene lacking the putative *SUF* promoter and the TSS (*g*Mp*FGMYB-S* [Y]; Fig 6C) accumulated Mp*FGMYB* transcripts to

the level comparable to or higher than that of wild-type females (Fig 6D) and morphologically converted to females (Fig 6E and F). In contrast, genetically male plants harboring the entire Mp*FGMYB/SUF* locus as a transgene, i.e., containing the putative *SUF* promoter and the TSS (*g*Mp*FGMYB-L*; Fig 6C), did not accumulate Mp*FGMYB* transcripts to the level comparable to wild-type females (Fig 6D) and retained male morphologies (Fig 6G and H). These observations indicate a key role of *SUF* transcription in suppressing Mp*FGMYB* at the same locus in males, and inability of the endogenous *SUF* locus to suppress unlinked transgenic copies of Mp*FGMYB*. Taken together, these data indicate that *SUF*-mediated Mp*FGMYB* suppression is locus-specific and hence acts *in cis*.

## Discussion

In most multicellular eukaryotes, sexual differentiation initially takes place in diploid reproductive organs that later produce, via meiosis, haploid gametes with the diploid body remaining functional and supporting fertilization and embryogenesis in case of placental organisms. Accordingly, sexual differentiation in land plants has been mainly studied from the perspective of the development of the diploid sporophytic organs, such as pistils and stamens of angiosperms. In contrast with the elaborate sexual morphologies of sporophytic organs, female and male gametophytes of

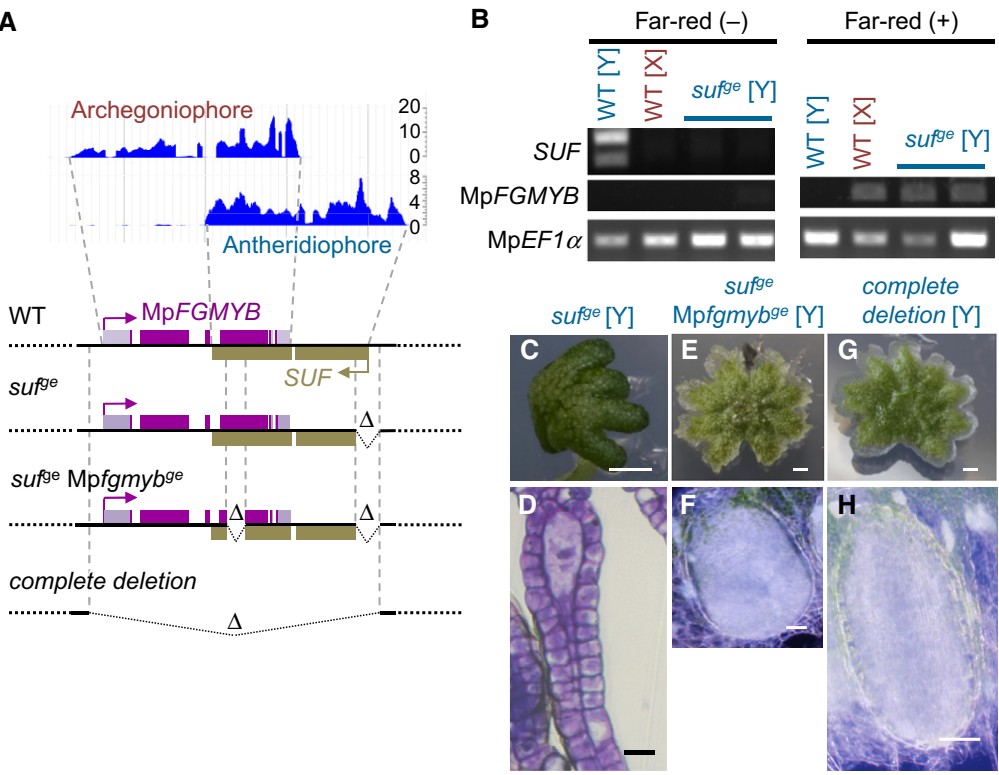

**Figure 5.  Antisense *SUF* suppresses Mp*FGMYB* expression in males.**

A    RNA-seq analysis showing male-specific accumulation of lncRNAs derived from the Mp*FGMYB* 3′ region (top), and diagrams illustrating wild-type and mutant Mp*FGMYB/SUF* loci (bottom). Folded lines with a Δ symbol indicate a deletion.

B    RT–PCR analysis of wild-type and genetically male *suf* mutants revealed loss of *SUF* expression and gain of Mp*FGMYB* expression in *suf* mutants after induction of reproductive growth by far-red irradiation. Two independent *suf* mutant alleles were analyzed. The *SUF* primer pair used here flanked an intron and the duplicated bands of *SUF* likely represent spliced and unspliced forms.

C–H  Gametangiophore morphology (C, E and G) and gametangium development (D, F and H) of plants with the designated genotypes. Scale bars, 1 mm (C, E and G), 20 μm (D), 50 μm (F), 100 μm (H).

Source data are available online for this figure.

angiosperms are reduced to seven-celled embryo sacs and three-celled pollen grains, respectively, making it difficult to study gametophytic sexual differentiation controls central to gametogenesis. A recently revived model bryophyte, *M. polymorpha,* exhibits conspicuous sexual dimorphism in their haploid-dominant growth phase and thus provides a unique opportunity to study sexual differentiation in gametophytes.

In this study, an autosomal Mp*FGMYB* was identified as a gene specifically expressed in the female sexual organ of *M. polymorpha* (Fig 2), and loss-of-function Mp*fgmyb* mutants exhibited a nearly complete female-to-male sex conversion phenotype (Figs 3 and 4). Furthermore, antisense transcription of Mp*FGMYB* occurs specifically in males, and disruption of this antisense gene, *SUF*, led to misexpression of Mp*FGMYB* and acquisition of nearly complete female morphology in males (Fig 5). Classical genetic studies suggested a female-dominant mode of sex determination system in *M. polymorpha* (reviewed in Bowman, 2016). Based on this notion, an attractive hypothesis is that expression of *SUF* is suppressed by a presumptive sex determinant (*Feminizer*) encoded by the X chromosome (Fig 7B; Haupt, 1932). In the absence of *Feminizer*, *SUF*

suppresses Mp*FGMYB* expression, allowing a default male differentiation program to proceed (Fig 7A).

Combined with the previously identified functions of FGMYB genes in female gametophyte development and synergid functions in *A. thaliana* (Fig 7C; Kasahara *et al*, 2005; Punwani *et al*, 2007; Rabiger & Drews, 2013), our study revealed that closely related R2R3 MYB-type transcription factors regulate female gametophyte development across the land plant lineage. Importantly, FGMYBs function during the haploid phase of the life cycle in both *A. thaliana* and *M. polymorpha*, implying that this was also the condition in the ancestral land plant. Thus, the ancient female gametophyte-promoting functions of FGMYBs appear to be retained in the embryo sac, the highly reduced female gametophyte of flowering plants (Fig 7C), while remaining responsible for the conspicuous sexual morphologies in the gametophytes of extant liverworts (Fig 7B). To what extent FGMYBs regulate conserved sets of target genes across the land plant lineage, however, remains an open question, as female gametophytes have been considerably diversified in morphology, while retaining their central function, egg cell production, in the course of land plant evolution.

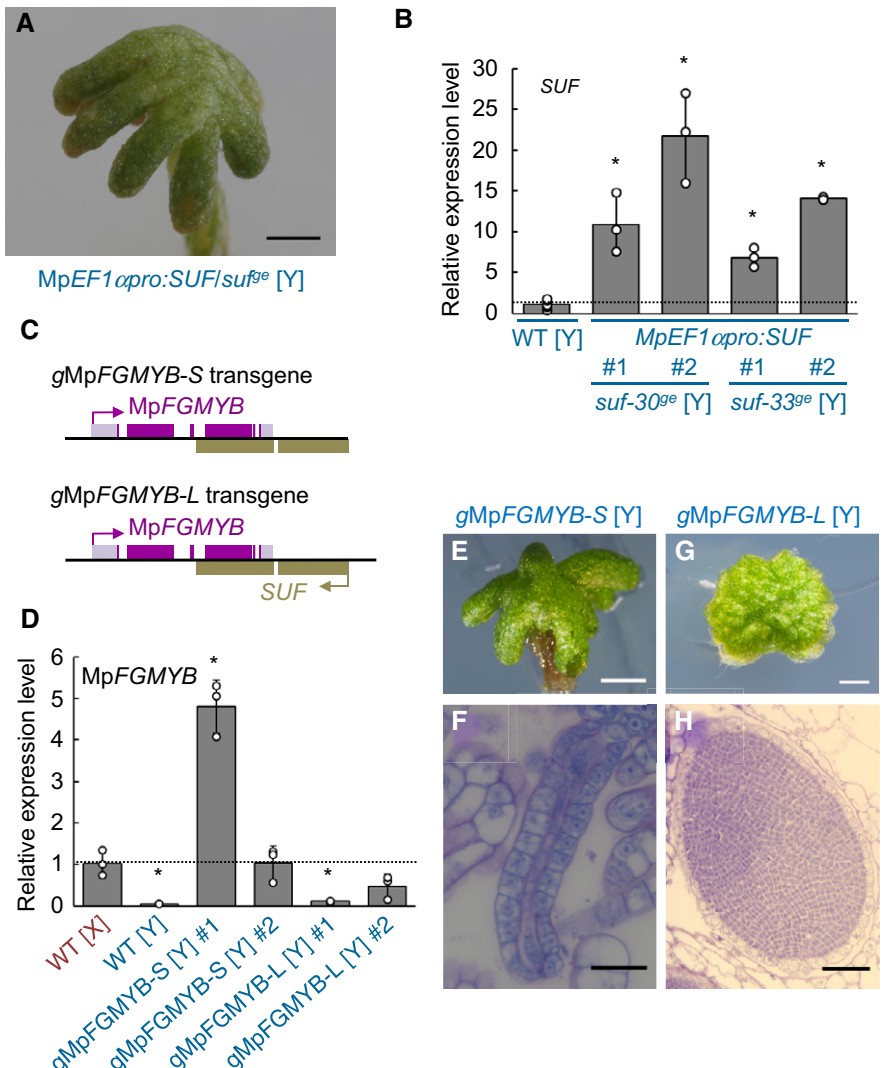

**Figure 6.  SUF acts *in cis*.**

A    A gametangiophore of Mp*EF1αpro:SUF/suf-30^ge* [Y], indicating the inability of transgenic *SUF* overexpression in rescuing the feminization phenotype of *suf^ge* [Y]. Scale bar, 2 mm.

B    Real-time RT–PCR analyses confirming *SUF* transcript accumulation in *SUF*-overexpressing lines. Constitutively expressed Mp*EF1α* was used as a control. Measurements of six biological replicates for WT [Y] and three biological replicates for each *SUF*-overexpressing line are plotted.

C    Structures of the *g*Mp*FGMYB-S* and *g*Mp*FGMYB-L* transgenes without (−S) or with (−L) the putative promoter and the TSS of *SUF*.

D    Real-time RT–PCR measurement of Mp*FGMYB* transcript levels. Three biological replicates are analyzed for each line.

E–H    Gametangiophores (E, G) and gametangia (F, H) of *g*Mp*FGMYB-S* [Y] (E, F) and *g*Mp*FGMYB-L* [Y] (G, H). Scale bars, 1 mm (E, G), 50 μm (F), 100 μm (H).

Data information: In (B) and (D), bars represent mean ± SD. Asterisks indicate significant differences from WT [Y] (B) or from WT [X] (D) ($P < 0.05$, two-tailed Student's *t*-test). See Source Data online for measurements and statistics.

Source data are available online for this figure.

It should be also noted that among the additional four *A. thaliana* genes in the FGMYB clade (Fig 2A), At*MYB115* and At*MYB118* are known to activate fatty acid synthesis in the endosperm (Troncoso-Ponce *et al*, 2016). Thus, at least some FGMYB homologs function transiently after fertilization. Considering the expression of Mp*FGMYB* in the sporophytes of *M. polymorpha* (Fig 2D), it is also possible that Mp*FGMYB* functions are also not limited to the gametophytes. As Mp*fgmyb* mutants are infertile, investigation of its sporophytic functions requires elaborate transgenic plants where Mp*FGMYB* expression can be temporally controlled. Alternatively, the sporophytic function

of AtMYB115 and AtMYB118 may have been acquired by evolutionary cooption of originally gametophytic regulators to the sporophytic functions, as we have previously proposed for the RKD genes family (Koi *et al*, 2016). In our RT–PCR data, Mp*FGMYB* and *SUF* are apparently coexpressed in the sporophytes (Figs 2D and EV4A). A trivial interpretation of this would be a relaxed suppression reflecting little or no functional roles of MpFGMYB in the sporophytes. Alternatively, this may reflect requirement of Mp*FGMYB* functions in limited time and/or space in the sporophytes as the *SUF*-mediated suppression should be cell-specific.

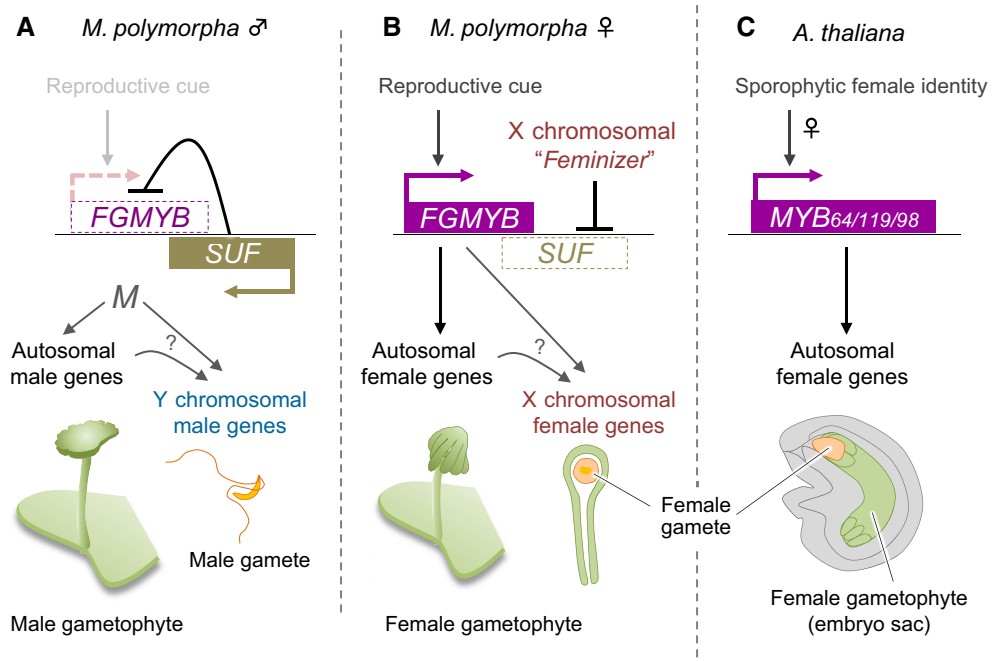

**Figure 7.  FGMYBs promote female gametophyte development in both *Marchantia polymorpha* and *Arabidopsis thaliana*, but with distinct upstream regulation.**

A, B   A bidirectional transcription module at the Mp*FGMYB/SUF* locus acts as a toggle switch between male (A) and female (B) sexual differentiation in *M. polymorpha*. Mp*FGMYB* expression is activated by one or more unknown cues associated with reproductive growth. In males, Mp*FGMYB* expression is suppressed by constitutively expressed antisense gene, *SUF*, allowing an unknown factor (*M*) to activate both autosomal and Y chromosomal genes to promote male differentiation (A). A dominant "*Feminizer*" on the X chromosome (Haupt, 1932; Bowman *et al*, 2017) directly or indirectly suppresses *SUF* expression, allowing expression of Mp*FGMYB* and downstream autosomal and X chromosomal genes to promote female differentiation (B). Genes on the X and Y chromosomes are dispensable for the sexual morphologies of the gametophytes, but required for the differentiation of functional gametes.

C      In *A. thaliana*, three FGMYB genes promote female differentiation in the embryo sac, a highly reduced female gametophyte of flowering plants. FGMYB expression is regulated at the transcriptional level after the formation of sporophytic female floral organs (Kasahara *et al*, 2005; Rabiger & Drews, 2013).

In animals, a plethora of sex determination pathways converge on a set of key regulators of sexual differentiation (Gamble & Zarkower, 2012; Bachtrog *et al*, 2014). Similarly, in this study we demonstrate that members of the FGMYB subfamily are regulators of female gametophyte development in two divergent lineages spanning the phylogenetic diversity of land plants, but that FGMYB regulation is under control of different sex determination mechanisms. In *M. polymorpha*, Mp*FGMYB* expression is downstream of a chromosomal sex determination locus in the free-living gametophyte generation. In contrast, in the derived gametophyte of flowering plants, FGMYB has been placed downstream of sporophytic sexual differentiation programs upon acquisition of the diploid-dominant life style in the lineage leading to the extant seed plants (Fig 7C). Identification of upstream factors activating FGMYB expression in *A. thaliana*, as well as comparative studies along the land plant lineage between bryophytes and flowering plants, will help elucidate how gametophytic sexual differentiation programs were incorporated into the regulatory networks governing sporophytic reproductive development.

Our genetic analyses also revealed that *SUF* can suppress Mp*FGMYB* expression only *in cis*. Although lncRNA-mediated regulation of gene expression has been reported for a large variety of physiological and developmental processes in plants, as in fungi,

yeasts, and animals (see Kopp & Mendell, 2018; Liu *et al*, 2015 for review), only in few cases are plant lncRNAs shown to act *in cis*. In the case of vernalization-induced suppression of *FLOWERING LOCUS C* (*FLC*) in *A. thaliana*, the antisense lncRNA COOLAIR initially forms dense clouds at the locus from which it is transcribed, and this leads to changes in epigenetic state to stabilize suppression (Csorba *et al*, 2014; Rosa *et al*, 2016; Yuan *et al*, 2016). Similarly, expression of *DELAY OF GERMINATION1* (*DOG1*) is suppressed during seed maturation in *A. thaliana* by its antisense RNA *asDOG1* in an allele-specific manner (Fedak *et al*, 2016). Although *SUF* shares many similarities with *COOLAIR* and *asDOG1*, such as defined molecular organization and *cis*-acting mode of suppression, molecular mechanisms by which the *SUF* lncRNAs (or *SUF* transcription *per se*) suppresses Mp*FGMYB* expression are yet to be investigated.

# Materials and Methods

### Phylogenetic analysis

Amino acid sequences of the R2R3-type MYB proteins in clades 11, 12, and 14–16 (Bowman *et al*, 2017) were used in the phylogenetic

analysis. These sequences were retrieved using the MarpolBase (http://marchantia.info/), Phytozome (https://phytozome.jgi.doe.gov/), and TAIR (https://www.arabidopsis.org/) databases (Rensing *et al*, 2008; Dubos *et al*, 2010; Bowman *et al*, 2017). Regions encompassing the ~110-residue MYB domain were aligned using the MUSCLE program (Edgar, 2004) in AliView v1.18.1 (Larsson, 2014). After manually removing the gaps, a phylogenetic tree was constructed using the maximum-likelihood algorithm based on the Le_Gascuel_2008 model (Felsenstein, 1981; Le & Gascuel, 2008) and evaluated using a bootstrap method (Felsenstein, 1985) with 1000 replicates in MEGA7 (Kumar *et al*, 2016).

## Plant materials

Male and female accessions of *M. polymorpha*, L. subsp. *ruderalis*, Takaragaike-1 (Tak-1), and Takaragaike-2 (Tak-2; Ishizaki *et al*, 2016), respectively, were cultured on half-strength Gamborg's B5 medium solidified with 1% (w/v) agar under continuous white light at 22°C. Plants were maintained asexually and propagated through the gemma. To induce reproductive development, 10-day-old thalli were transferred to half-strength Gamborg's B5 medium containing 1% (w/v) sucrose solidified with 1.4% (w/v) agar and illuminated with far-red LED lamps (VBL-TFL600-IR730; IPROS Co., Tokyo, Japan) in addition to white light.

*Arabidopsis thaliana* (L.) Heynh accession Col-0 was used as the wild type. The *AtMYB64pro:NYG* transcriptional reporter line was described previously (Waki *et al*, 2013). Seeds of *myb64-4* (SAIL_876_E05) and *myb119-3* (SALK_120501) were obtained from the Arabidopsis Biological Resource Center (Columbus, OH) and backcrossed twice with wild-type plants. The *gAtMYB64-Citrine* translational reporter lines were generated by introducing the pBIN41-gAtMYB64-Citrine construct described in Appendix Supplementary Methods into the plants homozygous for the *myb64-4* and heterozygous for the *myb119-3* alleles.

### DNA construction

Plasmids used in this study were constructed using gateway cloning system (Ishizaki *et al*, 2015) or SLiCE method (Motohashi, 2015). Primers used for DNA construction are listed in Table EV2. See Appendix Supplementary Methods for details.

### Generation of transgenic *Marchantia polymorpha*

The genome editing constructs pMpGE010_MpFGMYBge01, pMpGE010_MpFGMYBge02, pMpGE018_SUFge, and pMpGE018_-complete-deletion (see Appendix Supplementary Methods for details) were introduced into *M. polymorpha* as described previously (Ishizaki *et al*, 2008). Other constructs were introduced essentially as described previously (Kubota *et al*, 2013). For mutant isolation and sex diagnosis, gRNA-targeted regions of Mp*FGMYB* were amplified from the genomic DNAs prepared from the thalli of T1 plants using the primer pairs ge01-Fw/ge01-Rv and ge02Fw/ge02-Rv for MpFGMYBge01 and MpFGMYBge02, respectively. PCR products were directly sequenced using Bigdye Terminator v3.1 (Thermo Fisher Scientific) and the primers ge01-Fw or ge02-Fw. The genetic sex of the Mp*fgmyb*$^{ge}$ lines was diagnosed as described

previously (Fujisawa *et al*, 2001) using the primer sets listed in Table EV2.

### Histology and microscopy

Excised tissues were fixed with a formaldehyde/acetic acid (FAA) solution overnight at 4°C, and then dehydrated in an ethanol series and embedded in Technovit 7100 resin (Heraeus Kulzer, Hanau, Germany). Sections of 2 μm thickness were made using a Leica RM2155 microtome (Leica Microsystems, Wetzlar, Germany) and stained with toluidine blue. Image contrast was enhanced equally over the entire area using the Fiji program (Schindelin *et al*, 2012).

Confocal laser scanning microscopy was carried out using a Nikon C2 confocal laser scanning microscope (Nikon Instech, Tokyo, Japan). Archegonia were stained with 0.4% (v/v) SCRI Renaissance 2200 (Renaissance Chemicals, Selby, UK; Musielak *et al*, 2015) in 4% (w/v) paraformaldehyde and 1× PBS. Sperm released from antheridiophores were collected by centrifugation and stained with 1 μg/ml DAPI in 4% (w/v) paraformaldehyde and 1× PBS. Stained sperm were dropped on a slide glass and dried up by incubating for a few minutes at room temperature, then observed using a confocal microscope. The sperm attraction assay was performed as described previously (Koi *et al*, 2016).

For electron microscopy, the excised tissues were transferred to vials containing a fixative solution composed of 4% (v/v) glutaraldehyde and 0.05 M phosphate buffer (pH 7.0), and then vacuum-infiltrated until the specimens sank to the bottom. After fixation for 6 h at room temperature, samples were rinsed with 0.05 M phosphate buffer and post-fixed in 1% (w/v) osmium tetroxide for 3 h at 4°C. The samples were dehydrated in a graded ethanol series and embedded in Spurr's plastic resin using a graded series of propylene oxide and the resin. Ultra-thin sections (80 nm) were prepared with a diamond knife on an ultramicrotome (Ultracut R; Leica Microsystems). Sections were stained sequentially with uranyl acetate and lead citrate, and then observed under a transmission electron microscope (JEM 1010; JEOL, Tokyo, Japan).

## Data availability

RNA-seq datasets used in this study are deposited in the DDBJ Sequence Read Archive (https://www.ddbj.nig.ac.jp/dra/index-e.html) with the accession number DRA006846.

**Expanded View** for this article is available online.

### Acknowledgements
We thank Masako Kanda for providing technical assistance, Dr. Keisuke Inoue for providing the CRISPR/Cas (D10A) vectors, and Dr. Shunsuke Miyashima and Dr. Ryosuke Sano for helpful discussion. This work was supported by MEXT KAKENHI grants 17J08430 to TH, 25113007 to KN, 17H05841 and 18K06285 to SY, 25113009 and 17H07424 to TKo, and an Australian Research Council grant DP170100049 to JLB. TH was supported by a JSPS Fellowship for Young Scientists.

### Author contributions
TH, KO, MS, SY, TKa, KTY, and KN performed experiments and/or omics data analyses. TH, KO, SY, RN, KTY, JLB, TKo, and KN designed the project. TH, JLB

and KN wrote the manuscript. All authors jointly interpreted the data and thoroughly checked the manuscript.

## Conflict of interest

The authors declare that they have no conflict of interest.

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
