## [Review Process File · The EMBO Journal]

A *cis*-acting bidirectional transcription switch controls sexual dimorphism in the liverwort

Tetsuya Hisanaga, Keitaro Okahashi, Shohei Yamaoka, Tomoaki Kajiwara, Ryuichi Nishihama, Masaki Shimamura, Katsuyuki T. Yamato, John L. Bowman, Takayuki Kohchi and Keiji Nakajima.

Review timeline:

Submission date:	11 th July 2018
Editorial Decision:	13 th August 2018
Revision received:	17 th November 2018
Accepted:	26 th November 2018

Editor: Anne Nielsen

Transaction Report:

1st Editorial Decision

13th August 2018

Thank you for submitting your manuscript for consideration by the EMBO Journal. It has now been seen by three referees whose comments are shown below.

As you will see from the reports, all three referees express interest in the findings reported in your manuscript although they also raise a number of concerns that you will have to address before they can support publication of the manuscript here. Most importantly, refs #2 and #3 ask for additional characterization of the SUF RNA transcript structure, expression pattern and mechanism of action. I understand that a full mechanistic delineation is beyond the scope of the current study but I would strongly suggest you to follow ref #2's recommendation to test for a *cis* vs *trans* effect. Regarding point 4 from this reviewer, I would encourage you to run the computational analysis suggested but establishing evidence for conservation of this regulatory mechanism will not be a requirement for publication here. Ref #3 raises similar concerns as ref #2 and in addition stresses that the RNAseq data will need to be provided (preferably with an accession number available to the referees as also described in our author guidelines) and that the phylogenetic analysis for MpFGMYB should be extended.

Given the referees' positive recommendations, I would like to invite you to submit a revised version of the manuscript, addressing the comments of all three reviewers.

REFeree REPORTS.

Referee #1:

This manuscript describes the function of a *Marchantia polymorpha* locus that controls sexual development. The MpFGMYB transcription factor promotes female development, and an antisense transcript expressed, MpSUF, in males represses the expression of the MYB transcription factor.

The most important data in the manuscript are:

MpFGMYB is expressed in developing archaegoniophores (female, egg-producing structures) and not detectable in antheridiophores (male sperm producing structures). Genetically female Mpfgmyb loss of function mutants develop as males and produce sperm. However, these sperms are morphologically defective. This suggests that genes on the Y chromosome are required for sperm development.

MpSUF is expressed in males. MpSUF is an antisense MpFGMYB transcript and promotes maleness in male plants. Genetically male Mpsuf mutants are feminized; they produced archaegoniophores.

The arabidopsis genes that are most similar to MpFGMYB are also involved in female gametophyte (megagametophyte) development. This suggests that the same molecular mechanism controls the development of female gametophyte structures in these diverged groups of plants. This suggests that the FGMYB functioned in the last common ancestor of Arabidopsis and Marchantia.

This is a complete study and all relevant data are presented. I have no additions to suggest.

Referee #2:

This manuscript by Hisanaga et al describes the identification of a conserved MYB transcription factor (FGMYB) in the liverwort Marchantia that controls female gametophyte development. Loss of this gene results in female-to-male sex conversion. The gene is evolutionarily conserved also in higher plants. The authors go on to show that expression of FGMYB is regulated by a long-non-coding antisense RNA called SUF, which is upregulated in male gametophytes. Loss of SUF function results in a male-to-female sex conversion. The manuscript is very clearly written and the data support well the conclusions. Functional genomic analysis of developmental pathways in lower plants such as liverworts is a very interesting area that will provide important insights into the evolutionary dynamics of plant development. This study provides significant advances to our understanding of female gametophyte determination. To further strengthen the manuscript, I have a few suggestions for improvement.

1) The mutually exclusive transcription of FGMYB and SUF provides a tentative mechanism that is very appealing. However, besides the transcriptional analysis and the SUF deletion the manuscript lacks mechanistic insight into how this is happening. It would strengthen the manuscript considerably to dig a little deeper into this. With the current evidence, it is possible that either SUF RNA inhibits FGMYB transcription (action in trans) or that transcription of the SUF locus per se inhibits FGMYB transcription (action in cis). As a strain with a deletion of SUF is already available, it should be relatively easy to test whether supplying SUF RNA in trans (from a transgene) can complement the suf deletion phenotype. This would provide important evidence whether SUF acts in cis or in trans.

2) It would also be helpful to characterize the SUF transcript a bit more to confirm and complement the RNA-seq data. Where does it start, where does it end, does it have introns etc.? The RT-PCR confirmation provided for SUF (Fig. 5b and Fig. S6) is confusing as one Figure shows a double band and the other does not. Could the authors explain this double band and provide RT-PCR evidence for the full-length SUF transcript?

3) The authors have previously described RKD as a crucial regulator of egg cell maturation in Marchantia (Koi, 2016). As RKD appears as a possible target of FGMYB, the authors should check expression of RKD in fgmyb and suf mutant strains.

4) Given that FGMYB is conserved in higher plants such as *A. thaliana*, it would be interesting to see whether the regulation through an antisense RNA is also likely conserved. Is there any evidence that any of the Arabidopsis orthologues also have a SUF-like antisense RNA (not necessarily related in sequence)? Addressing this (even if only by carefully analysing available and published transcriptome data) would strengthen the role of SUF in the regulation of FGMYB.

Small comments:

- p. 9 "shorter nuclei" - unclear what is meant by this. Please clarify.

- Fig. 4a-c: Please clarify what can be seen here by indicating the nucleus etc.
- Fig. 5a: The scales of the reads of sense and antisense are vastly different (20 vs. 8), yet the font of the scale is so small, that it is hard to appreciate this. While it may not be practical to show everything at the same scale, a simple solution would be to make the y axis labelling larger so that this difference could be seen more clearly.

Referee #3:

The Authors have identified a sex determining factor in *Marchantia*, the new model to study evolution of land plants. Comparing transcriptomes between vegetative haploid tissues and the female reproductive organs (archegonia), a MYB TF is identified, which is named FEMALE GAMETOHYTIC SPECIFIC MYB (MpFGMYB). This TF belongs to a small clade of MYB genes. The Arabidopsis genes from this clade are also expressed in the female gametophyte of *At*. KO mutant of this gene in female plants results in a dramatic conversion of sexual organ development into male organs - antheridiophores-. This conversion is marked by the change of morphology and the striking development of male gametes instead of female gametes on female mutant KO plants. The phenotype is complemented by expression of WT as well as the GFP- fusion protein used as reporter. However, sperm showed defect in flagella development and motility that could be the result of the lack of genes on the Y chromosome that is obviously absent from these mutant female plants. The study proceeds with demonstration that MpFGMYB expression is suppressed by an antisense SUPPRESSOR OF FEMINIZATION (SUF) that is expressed only in male gametophytes after induction of sexual reproduction by FR. The KO of SUF in male results in production of female looking reproductive organs that differentiate archegoniophores. Deletion of the entire locus (absence of both sense MpFGMYB and SUF) resulted in WT-looking male reproductive organs with fertile sperm. Overall experiments are conducted correctly and the results obtained are unambiguous. These first striking findings open the path to our understanding of the origin of sexual differentiation in land plants. In addition the study provides a valuable model in addition to FLC in angiosperms to study the bases of transcriptional regulation by antisense lnc RNAs. As such the results of this work are of interest to a broad readership.

Major points

1. study is based on the analysis of transcriptomes of archegonia and thalli which is not provided by the Authors. It is essential and conditional to publication that this dataset is shared in a public database and Table of the results provided in supplementary materials.
2. The specific expression of MpFGMYB and SUF are the cornerstone of this study. So the Authors provide data more extensive and convincing than a simple RT-PCR. For example a Quantitative RT-PCR survey of expression of MpFGMYB and SUF across different stages of vegetative and reproductive development of male and female gametophytes and in the sporophyte. This should also include different stages of antheridiophore and archegoniophore development after induction by FR light, including times when the organogenesis has not started. Given the function of the gene, one expects an expression that pre-dates the morphological development of the archegoniophore.
3. The authors highlight a conserved domain present in MpFGMYB and its putative orthologs in the moss *Physcomitrella*. Could the Authors provide a more extensive search for potential orthologs amongst bryophytes to ascertain the nature of this motif. At least the Authors should provide a basal study of expression of the potential orthologs of MpFGMYB in *Physcomitrella* that does not show reproductive organ differentiation but only production of antheridia and archegonia on the same plant. Could the Authors tell us at least if an ortholog of SUF exists in *Physcomitrella*?

Minor points

1. According to the phylogenetic analysis there are four other MYB genes in *At* in the same clade as MpFGMYB. Could the authors comment on their expression and if they are not expressed in the female gametophyte, what could differentiate these genes?
2. In the discussion the Authors point out that RKD might be regulated by MpFGMYB. Again, either the Authors attempt to obtain data related to this question and discuss the result. This would have been quite easy. Or they can remove this point of discussion.

3. In the discussion there are statement regarding potential parallels in the role of putative orthologs of MpFGMYBs in At. These should be toned down very much or rather removed because there is no data besides the expression of these putative orthologs in the embryo sac, to propose a potential orthology. The At protein does not show the same N-terminal domain conserved amongst bryophyte. If the Authors wish to discuss this point they should attempt cross-complementation of Mp mutant with the At gene. Or at least an engineered fusion between the At MYB domain replacing the Mp MYB domain of MpFGMYB.
4. In the Introduction the concept of definition of sexual organ identity is confused with the definition of gamete differentiation at several points. This should not be the case and the Authors should rigorously compare what is truly comparable between angiosperms and bryophytes.
5. In the abstract the statement that bidirectional transcription is the key mechanism should be toned down because it is not shown in this study whether the Antisense SUF plays a role directly in the repression of MpFGMYB.
6. The last sentence of the Abstract is not substantiated by findings presented here and should be modified.

1st Revision - authors' response

17th November 2018

Response to the referees' comments

Referee #1:

This manuscript describes the function of a *Marchantia polymorpha* locus that controls sexual development. The MpFGMYB transcription factor promotes female development, and an antisense transcript expressed, MpSUF, in males represses the expression of the MYB transcription factor.

The most important data in the manuscript are:

MpFGMYB is expressed in developing archaegoniophores (female, egg-producing structures) and not detectable in antheridiophores (male sperm producing structures). Genetically female *Mpfgmyb* loss of function mutants develop as males and produce sperm. However, these sperms are morphologically defective. This suggests that genes on the Y chromosome are required for sperm development.

MpSUF is expressed in males. MpSUF is an antisense MpFGMYB transcript and promotes maleness in male plants. Genetically male *Mpsuf* mutants are feminized; they produced archaegoniophores.

The arabidopsis genes that are most similar to MpFGMYB are also involved in female gametophyte (megagametophyte) development. This suggests that the same molecular mechanism controls the development of female gametophyte structures in these diverged groups of plants. This suggests that the FGMYB functioned in the last common ancestor of Arabidopsis and Marchantia.

This is a complete study and all relevant data are presented. I have no additions to suggest.

We thank the referee for his/her highly favorable comments on our study.

Referee #2:

This manuscript by Hisanaga et al describes the identification of a conserved MYB transcription factor (FGMYB) in the liverwort *Marchantia* that controls female gametophyte development. Loss of this gene results in female-to-male sex conversion. The gene is evolutionarily conserved also in higher plants. The authors go on to show that expression of FGMYB is regulated by a long-non-coding antisense RNA called SUF,

which is upregulated in male gametophytes. Loss of *SUF* function results in a male-to-female sex conversion. The manuscript is very clearly written and the data support well the conclusions. Functional genomic analysis of developmental pathways in lower plants such as liverworts is a very interesting area that will provide important insights into the evolutionary dynamics of plant development. This study provides significant advances to our understanding of female gametophyte determination. To further strengthen the manuscript, I have a few suggestions for improvement.

We thank the referee for his/her constructive comments on our paper. Our answers to the suggestions are as follows.

1) The mutually exclusive transcription of *FGMYB* and *SUF* provides a tentative mechanism that is very appealing. However, besides the transcriptional analysis and the *SUF* deletion the manuscript lacks mechanistic insight into how this is happening. It would strengthen the manuscript considerably. With the current evidence, it is possible that either *SUF* RNA inhibits *FGMYB* transcription (action in trans) or that transcription of the *SUF* locus per se inhibits *FGMYB* transcription (action in cis). As a strain with a deletion of *SUF* is already available, it should be relatively easy to test whether supplying *SUF* RNA in trans (from a transgene) can complement the *suf* deletion phenotype. This would provide important evidence whether *SUF* acts in cis or in trans.

While the first version of this manuscript was under review, we obtained new data indicating a *cis*-acting mode of *SUF* on *MpFGMYB* suppression. We originally intended to use these data for separate publication in future, but now decided to incorporate them to the present paper. As suggested by the referee, we introduced *SUF* overexpression constructs into *suf* [Y] mutants and found no complementation despite high levels of *SUF* transcripts accumulation. Additionally, we introduced genomic fragment of the *MpFGMYB/SUF* locus with and without the putative *SUF* promoter and TSS (*gMpFGMYB-L* and *gMpFGMYB-S*, respectively) into WT [Y]. While *gMpFGMYB-S* with no predicted *SUF* expression could confer partial sex conversion to females, *gMpFGMYB-L* with predicted *SUF* expression could not. These data indicate the inability of transgenic *SUF* copies to suppress endogenous *MpFGMYB*, as well as of endogenous *SUF* to suppress transgenic *MpFGMYB* copies. Based on these observations, we concluded that *SUF* acts *in cis* to suppress *MpFGMYB* expression in males. We added these new findings as Fig 6 and described them in the last part of the Results section and also in the Discussion. The title for the paper has been changed to feature this important new finding.

2) It would also be helpful to characterize the *SUF* transcript a bit more to confirm and complement the RNA-seq data. Where does it start, where does it end, does it have introns etc.? The RT-PCR confirmation provided for *SUF* (Fig. 5b and Fig. S6) is confusing as one Figure shows a double band and the other does not. Could the authors explain this double band and provide RT-PCR evidence for the full-length *SUF* transcript?

We admit that characterization of *SUF* transcript structures is important, and thus performed 3' and 5' RACE PCRs for *SUF*. The data indicate that *SUF* transcripts have a well-defined structure with an invariable 5' end point and polyadenylation site, indicating that *SUF* represents a strictly defined transcription unit. We added these data in Appendix Fig S3. As for the double band, we apologize for the confusing figures. For RT-PCR in Fig 5b, we used a primer pair flanking the *SUF* intron, but the primers used in Fig S6a (currently Fig EV4B) do not. Since total RNA samples were treated with DNase I and the bands appeared specifically in wild-type males, we think the doubled band in Fig 5B to be derived from spliced (lower band) and unspliced (upper band) variants of *SUF* transcripts. We explained this in the legend to Fig 5. In vernalization-dependent suppression of *FLC*, unspliced *COOLAIR* lncRNA molecules are shown to form dense clouds at the locus from which they are produced leading to local change in epigenetic state (Rosa et al., Nat

Commun. 2016, 7: 13031). While it is hypothesized that unspliced *SUF* transcripts act similarly to unspliced *COOLAIR* lncRNA, it is still highly speculative and hence is not mentioned in the revised manuscript.

3) The authors have previously described RKD as a crucial regulator of egg cell maturation in *Marchantia* (Koi, 2016). As RKD appears as a possible target of GAMYB, the authors should check expression of RKD in *fgmyb* and *suf* mutant strains.

While we are interested in checking whether RKDs are transcriptional targets of FGMYBs, it would be difficult to test this possibility by simply examining expression of *MpRKD* in *Mpfgmyb* and *suf* mutants, because *MpRKD* expression is activated not only in female archegonia but also in male antheridia (Koi et al., 2016 Curr. Biol. 26: 1775-1781) likely by other regulators than *MpFGMYB*. In response to the referee #3's minor comment 2, we removed the part describing possible relationship between *MpFGMYB* and *MpRKD* from the Discussion.

4) Given that FGMYB is conserved in higher plants such as *A. thaliana*, it would be interesting to see whether the regulation through an antisense RNA is also likely conserved. Is there any evidence that any of the Arabidopsis orthologues also have a *SUF*-like antisense RNA (not necessarily related in sequence)? Addressing this (even if only by carefully analyzing available and published transcriptome data) would strengthen the role of *SUF* in the regulation of GAMYB.

We searched for publically available RNA-seq datasets from *Arabidopsis* pollen (Qin et al., 2018, F1000Res. 7: 93, doi:[10.12688/f1000research.13311.1]) and various tissues of *Physcomitrella patens* (Perroud et al., 2018, Plant J. 95: 168-182; Koshimizu et al, 2018, Nat Plants 4: 36–45). In the *Arabidopsis* pollen dataset, we did not detect antisense transcripts in none of the three *FGMYB* loci (*MYB64*, *MYB119* and *MYB98*), consistent with the lack of lncRNA-dependent regulation in *A. thaliana* male gametophytes discussed in the previous version (currently Fig 7C). We found that *FGMYB* homologues in *P. patens* are preferentially expressed in gametophores which include archegonia. We also detected a certain numbers of reads of the *FGMYBs* in antheridia. Although distribution patterns of reads along the loci were somewhat different between gametophores and antheridia, potentially suggesting accumulation of antisense lncRNAs, it is so far difficult to comment on possible occurrence of antisense transcription for *PpFGMYBs*, as these RNA-seq data are not from stranded libraries.

Small comments:

- p. 9 "shorter nuclei" - unclear what is meant by this. Please clarify.

We changed "shorter nuclei" to "incompletely condensed nuclei" (currently in p. 10).

- Fig. 4a-c: Please clarify what can be seen here by indicating the nucleus etc.

We now annotated the images with arrows and arrowheads which indicate flagella and nuclei, respectively.

- Fig. 5a: The scales of the reads of sense and antisense are vastly different (20 vs. 8), yet the font of the scale is so small, that it is hard to appreciate this. While it may not be practical to show everything at the same scale, a simple solution would be to make the y axis labelling larger so that this difference could be seen more clearly.

We now labeled the Y axis with larger fonts to present the data more clearly.

Referee #3:

The Authors have identified a sex determining factor in *Marchantia*, the new model to study evolution of land plants. Comparing transcriptomes between vegetative haploid tissues and the female reproductive organs (archegonia), a MYB TF is identified, which is named FEMALE GAMETOHYTIC SPECIFIC MYB (MpFGMYB). This TF belongs to a small clade of MYB genes. The Arabidopsis genes from this clade are also expressed in the female gametophyte of *At*. KO mutant of this gene in female plants results in a dramatic conversion of sexual organ development into male organs - antheridiophores-. This conversion is marked by the change of morphology and the striking development of male gametes instead of female gametes on female mutant KO plants. The phenotype is complemented by expression of WT as well as the GFP- fusion protein used as reporter. However, sperm showed defect in flagella development and motility that could be the result of the lack of genes on the Y chromosome that is obviously absent from these mutant female plants. The study proceeds with demonstration that MpFGMYB expression is suppressed by an antisense SUPPRESSOR OF FEMINIZATION (SUF) that is expressed only in male gametophytes after induction of sexual reproduction by FR. The KO of SUF in male results in production of female looking reproductive organs that differentiate archegoniophores. Deletion of the entire locus (absence of both sense MpFGMYB and SUF) resulted in WT-looking male reproductive organs with fertile sperm. Overall experiments are conducted correctly and the results obtained are unambiguous. These first striking findings open the path to our understanding of the origin of sexual differentiation in land plants. In addition the study provides a valuable model in addition to FLC in angiosperms to study the bases of transcriptional regulation by antisense lnc RNAs. As such the results of this work are of interest to a broad readership.

We thank the referee for his/her constructive comments on our paper. Our answers to the suggestions are as follows.

Major points

1. study is based on the analysis of transcriptomes of archegonia and thalli which is not provided by the Authors. It is essential and conditional to publication that this dataset is shared in a public database and provided in supplementary materials.

We thank the referee for pointing this out. We now deposited our transcriptome data to the DDBJ sequence Read Archive with an accession number of DRA006846. We added this information to the Materials and Methods section with a subheading Data availability. We also included the list of the 23 genes enriched in female gametophytes of both *M. polymorpha* and *A. thaliana* as Table EV1. When we were converting the transcript ID of the original dataset (CUFF.xxx) to the *Marchantia* gene ID (Mapolyxxxxsxxxx), we found that two of the selected transcripts are from an identical locus. Therefore, the number of selected genes has been reduced to 23.

2. The specific expression of MpFGMYB and SUF are the cornerstone of this study. So the Authors provide data more extensive and convincing than a simple RT-PCR. For example a Quantitative RT PRC survey of expression of MpFGMYB and SUF across different stages of vegetative and reproductive development of male and female gametophytes and in the sporophyte. This should also include different stages of antheridiophore and archegoniophore development after induction by FR light, including times when the organogenesis has not started. Given the function of the gene, one expects an expression that pre-dates the morphological development of the archegoniophore.

We appreciate the referee's constructive comments and carried out detailed expression analyses of MpFGMYB as shown in Fig 2D and Fig EV1 in the revised version. After confirming with real-time RT-PCR a steep rise of MpFGMYB expression levels upon induction of reproductive growth by FR irradiation (Fig. 2D), we used the

gMpFGMYBresist-Citrine (Fig EV1) to determine when and where the expression of *MpFGMYB* is initiated. Consistent with the predicted role of *MpFGMYB* in establishing female sexual morphologies, the expression of *MpFGMYB* was found to initiate around 10 days after the induction at the ventral side of the apical notch region (Fig EV1C) where archegoniophores later develop (Fig EV1D). As for *SUF* expression, our previous RT-PCR analyses (now shown in Fig EV4B) detected *SUF* transcripts both in vegetative thalli and antheridiophores. This was confirmed by real-time RT-PCR (Fig. EV4A). This analysis also revealed that antheridiophores exhibited considerably higher accumulation of *SUF* transcripts than thalli. Somewhat unexpectedly, *MpFGMYB* and *SUF* are coexpressed in the sporophytes. While our tentative interpretation for this is spatial or even temporal separation of *MpFGMYB* and *SUF* expression domains, this can not be resolved by currently available RT-PCR data obtained from bulked plant samples. We added the real-time RT-PCR data and corresponding discussion to the revised manuscript. As additional expression was detected in the sporophytes, name of the gene has been changed from "*FEMALE GAMETOPHYTE-SPECIFIC MYB*" to "*FEMALE GAMETOPHYTE MYB*".

3. The authors highlight a conserved domain present in *MpFGMYB* and its putative orthologs in the moss *Physcomitrella*. Could the Authors provide a more extensive search for potential orthologs amongst bryophytes to ascertain the nature of this motif. that does not show reproductive organ differentiation but only production of antheridia and archegonia on the same plant. Could the Authors tell us at least if an ortholog of *SUF* exists in *Physcomitrella*?

As we described in our answer to the referee #2's comment 4) above, we found that the *P. patens FGMYB* homologues are preferentially expressed in gametophores that include archegonia. We also detected a certain number of reads of *FGMYBs* in antheridia, and the distribution pattern of reads along the loci appeared different from those in gametophores. Although these differences suggest occurrence of antisense transcription at the *P. patens FGMYB* loci in antheridia, we think it still premature to mention this in the text, as the RNA-seq data are not derived from stranded libraries. As *P. patens* is monoicous and thus produces archegonia and antheridia in a close proximity in a single individual, more sophisticated experiments are required to unambiguously examine whether similar bidirectional transcription operates for spatiotemporal regulation of sexual morphologies in this bryophyte species. This will be done in future as a collaborative study with moss experts.

Minor points

1. According to the phylogenetic analysis there are four other MYB genes in *At* in the same clade as *MpFGMYB*. Could the authors comment on their expression and if they are not expressed in the female gametophyte, what could differentiate these genes?

We thank the referee for pointing out this important question. Among the four MYB genes, *MYB115* and *MYB118* are reported to be expressed in the endosperm and activate fatty acid synthesis, whereas *MYB22* and *MYB100* expression is generally quite low in various tissue types (Troncoso-Ponce et al., 2016 Plant Cell 28: 2666-2682). Thus at least some extant *FGMYB* members are expressed outside the female gametophyte, but rather function after fertilization. We speculate this to be a result of evolutionary cooption of gametophytic regulators to sporophytic functions, as has been proven the case for many developmental genes (see for example; Jang et al., 2011, Development 138: 2273-2281 and Kubota et al., 2014, Nat. Commun. 5: 3668), and also suggested for regulators of sexual reproduction by us (Koi et. al., 2016, Curr. Biol. 26: 1775-1781). We discussed this in the revised version.

2. In the discussion the Authors point out that *RKD* might be regulated by *MpFGMYB*. Again, either the Authors attempt to obtain data related to this question and discuss the result. This would have been quite easy. Or they can remove this point of discussion.

In response to this comment, we removed the corresponding paragraph from the Discussion.

3. In the discussion there are statement regarding potential parallels in the role of putative orthologs of MpFGMYBs in At. These should be toned down very much or rather removed because there is no data besides the expression of these putative ortholgs in the embryo sac, to propose a potential orthology. The At protein does not show the same N-terminal domain conserved amongst bryophyte. If the Authors wish to discuss this point they should attempt cross-complementation of Mp mutant with the At gene. Or at least an engineered fusion between the At MYB domain replacing the Mp MYB domain of MpFGMYB.

We agree with the referee; we think regulators for female sexual differentiation in gametophytic phase have common origin between Mp and At, but function of these regulators including their targets could be diverse. To make this point clear, we added a text into the Discussion as follows "To what extent FGMYBs regulate conserved sets of target genes across the land plant lineage, however, remains an open question, as female gametophytes have been considerably diversified in morphology, while retaining their central function, egg cell production, in the course of land plant evolution." A known role of some homologs in sporophytes is also mentioned in the following part in response to the minor point #1.

4. In the Introduction the concept of definition of sexual organ identity is confused with the definition of gamete differentiation at several points. This should not be the case and the Authors should rigorously compare what is truly comparable between angiosperms and bryophytes.

We apologize for the confusion between sexual organ identity and gamete differentiation in the Introduction. What we report here is regulation of gametophytic sexual differentiation processes in the liverwort and its comparison with the angiosperm counterpart, in particular developmental regulation of the female gametophytes, the embryo sacs. In the revised manuscript, sexual organ identity is mentioned only in the second paragraph where sex determination of floral organs is described.

5. In the abstract the statement that bidirectional transcription is the key mechanism should be toned down because it is not shown in this study whether the Antisense *SUF* plays a role directly in the repression of MpFGMYB.

As described in response to referee #2, results from additional experiments indicate that *SUF* acts *in cis*. This finding is included in the Abstract, and also discussed in the main text. We also changed the title of the paper featuring this important notion, rather than emphasizing conserved roles of the MYB subfamily in the previous one. This is also to respond to the minor point 3. However, as the referee pointed out, it is still difficult to conclude that the repression is truly at the transcription step. Therefore, in the Abstract, the sentence has been changed to "Thus the bidirectional transcription module at the MpFGMYB/*SUF* locus acts as a toggle between female and male sexual differentiation in *M. polymorpha* gametophytes". Similar expressions in the main text have been also revised accordingly.

6. The last sentence of the Abstract is not substantiated by findings presented here and should be modified.

In response to this comment, we changed the last sentence of the Abstract from "Thus, conserved regulators are inherently used in the female gametophytes development across

land plants but with diverse upstream regulation" to "Thus, phylogenetically related MYB transcription factors regulate female gametophyte development across land plants".

Accepted

26th November 2018

Thank you for submitting a revised version of your manuscript. It has now been seen by two of the original referees whose comments are shown below.

As you will see they both find that all criticisms have been sufficiently addressed and I am therefore pleased to inform you that your manuscript has now been accepted for publication in The EMBO Journal. Before we can send your files off to production I would ask you to address the following points

REFERE REPORTS

Referee #2:

In the revised version, Hisanaga et al. have made a very thorough attempt to address the concerns that were raised previously. The mode of action of SUF has been clarified and the additional data strengthen the conclusions considerably. I recommend acceptance of the revised manuscript for publication.

Minor comment: It would be helpful to add the number of clones that have been analyzed for the 5' and 3' RACE experiments, and the number of clones that support each position.

Referee #3:

The authors have addressed my comments adequately and the work is ready for publication

Corresponding Author Name: Keiji Nakajima

Manuscript Number: EMBOJ-2018-100240R